# Methodology for Detecting Progressive Damage in Structures Using Ultrasound-Guided Waves

**DOI:** 10.3390/s22041692

**Published:** 2022-02-21

**Authors:** Gerardo Aranguren, Javier Bilbao, Josu Etxaniz, José Miguel Gil-García, Carolina Rebollar

**Affiliations:** 1Department of Electronic Technology, Faculty of Engineering of Bilbao, University of the Basque Country (UPV/EHU), 48013 Bilbao, Spain; josu.etxaniz@ehu.es; 2Applied Mathematics Department, Faculty of Engineering of Bilbao, University of the Basque Country (UPV/EHU), 48013 Bilbao, Spain; javier.bilbao@ehu.es (J.B.); carolina.rebollar@ehu.es (C.R.); 3Department of Electronic Technology, Faculty of Engineering of Vitoria, University of the Basque Country (UPV/EHU), 01006 Vitoria-Gazteiz, Spain; jm.gil-garcia@ehu.es

**Keywords:** SHM, piezoelectric transducers, corrosion, pattern matching, pattern recognition

## Abstract

Damage detection in structural health monitoring of metallic or composite structures depends on several factors, including the sensor technology and the type of defect that is under the spotlight. Commercial devices generally used to obtain these data neither allow for their installation on board nor permit their scalability when several structures or sensors need to be monitored. This paper introduces self-developed equipment designed to create ultrasonic guided waves and a methodology for the detection of progressive damage, such as corrosion damage in aircraft structures, i.e., algorithms for monitoring such damage. To create slowly changing conditions, aluminum- and carbon-reinforced polymer plates were placed together with seawater to speed up the corrosion process. The setup was completed by an array of 10 piezoelectric transducers driven and sensed by a structural health monitoring ultrasonic system, which generated 100 waveforms per test. The hardware was able to pre-process the raw acquisition to minimize the transmitted data. The experiment was conducted over eight weeks. Three different processing stages were followed to extract information on the degree of corrosion: hardware algorithm, pattern matching, and pattern recognition. The proposed methodology allows for the detection of trends in the progressive degradation of structures.

## 1. Introduction

The implementation of structural health monitoring (SHM) techniques will be a challenge for aviation in the coming years [1]. This implementation increases safety, reduces maintenance costs, and redefines the design of aircraft structures. For more than two decades, there has been a growing interest in this field with contributions on sensors, testing techniques [2], and signal processing algorithms [3].

The SHM methods developed and their approach to the observation of the different damages in the structures are influenced, when not fully determined, by the features and type of the sensors those methods relay on [4]. Sensors are a key-element in SHM. In fact, they define the differences among SHM techniques. Sensing can be direct or indirect. There are sensors that suffer the damage at the same time as the structure does (direct sensing) [5]: fibers, strain gauges, etc. They can be used to monitor the damage in structures; however, their success is limited by the position in which they are located along the structure. Structures can also be monitored by using them as a means to transmit its state (indirect sensing) using eddy currents [6], ultrasounds [7], or vibrations [8]. Indirect sensing methods are more effective and easier to implement than direct-sensing methods because they do not require covering the whole structure. 

Despite that they are well-known sensors, piezoelectric wafer active sensors (PWAS) [9,10] are considered a hot topic of research. Ultrasound methods can be classified following several criteria. If modes of operation are considered, ultrasound methods can operate in active and passive modes [11]. The active mode or ultrasonic guided wave tests (UGWT) consist of the emission of ultrasound waveforms, the propagation along the structure under test, and the reception of the echoes of the waveforms. Passive mode consists of the reception of the ultrasound waveforms generated after any impact or similar cause. 

Aerospace applications are undergoing a transition from using metallic materials to composite materials to build structures—mainly carbon-fiber-reinforced polymers (CFRP) [12]. Metals provide higher electric and sound conductivity than do composite materials. As a result, acoustic wave propagation is less attenuated in metals than in composites and allows the monitoring of larger surfaces. Furthermore, visual inspection does not always provide satisfactory results on CFRP [13], such as when inner damage takes place. In recent years, this has led to a growing interest in the application of SHM to composite materials. Hence, SHM technologies must be effective on both metals and composite structures.

Several criteria are considered to classify structural damage. For instance, if the structural damage is classified according to the time it takes to occur, it can be either progressive damage or sudden damage. Sudden damage is due to impact, fiber breakage in composite material, or operation failure [14,15]. Sudden damage can be assessed listening to acoustic emissions—namely, with passive ultrasound techniques. On the other hand, progressive damage refers to corrosion [16], delamination [17], or fatigue [18]. It is usually scattered in the structure and is complex to represent mathematically. The monitoring of progressive damage using ultrasounds in a structure requires testing the structure with active techniques, such as UGWT over a sufficiently long time for the damage to appear.

In aerospace real-world applications, other constraints for SHM technology can be found. For example, an aircraft consists of hundreds of structures that should be monitored several times a day. In addition, the SHM equipment must be lightweight. The important effects that temperature has on the propagation of acoustic waves [19] must be kept in mind. Eventually, the maintenance staff must be provided with real-time information about damage on an aircraft for a detailed assessment of the damage and its repair. An SHM system is expected to quickly report changes in the structures under monitoring without any information overload.

The goal of this research is to introduce a series of procedures, equipment, and algorithms for monitoring progressive damage over the life of an aeronautic structure. To validate the concepts introduced here, two plates, one made of aluminum and the other one made of CFRP, were tested under corrosion for eight weeks. Given the complexity of the testing and the data processing procedure, the empirical method is considered to validate the progressive damage monitoring proposal raised in this paper.

Section 2 gives details about the materials and methods applied to this research. Section 3 explains the signal algorithm pre-process hardware stage of the methodology proposed in the paper. The pattern-matching process stage is introduced in Section 4, and the pattern-recognition process stage is in Section 5. The paper ends with a summary of our conclusions in Section 6. 

## 2. Materials and Methods

The challenge of this research is to determine a suitable method to detect progressive damage in structures using UGWT. Specifically, several hypothesis for this research are considered, i.e., the need to perform the tests over several days, the requirement of multiple PWAS, carrying out active tests, the utilization of dedicated equipment, and the analysis of the data with several types of algorithms. To test these hypotheses, the detection of corrosion in structures is considered as proof of concept. 

Corrosion is an example of progressive damage. It does not appear abruptly; therefore, it is necessary to carry out a test campaign long enough to notice the effects of the damage. The materials most commonly used in the construction of aeronautical structures are aluminum and CFRP. These materials are never put together, since aluminum in contact with CFRP produces a significant deterioration in both materials, especially in a marine environment [20]. However, since this paper shows a new methodology for detecting progressive damage, two structures made of the aforementioned materials were used because the corrosion that occurs and appears in a few weeks is a well-known phenomenon. They were kept together, one over the other with seawater in between. The specifications of the structures are the following:-A CFRP plate of 800 × 400 mm, cross-ply of [0_2_/90_4_]s (12 plies of 0.15 mm, total 1.8 mm thickness).-An aluminum plate of 1003 × 503 mm, QQA250/5 ‘O’ 2024 aeronautic grade with a thickness of 1 mm.

Ten PWAS were permanently bonded with epoxy on each plate. Each PWAS is 7 mm diameter and 0.2 mm thick (Steminc model SMD07T02R412WL). The PWAS were aligned, spaced 10 mm apart, parallel, and centered on the short side of the plate, 35 mm from the edge. The number of transducers was chosen to use the round-robin mode and to show the need for multiple tests. The 10 PWAS ensured that sufficient information was collected from the structure to detect the damage; however, it is likely that one could obtain the same results with a smaller number of PWAS. The distance between the transducers eases the use of a minimum wiring that can be integrated into a PCB [21], reducing the weight and the odds for breakage.

Even though currently some dedicated equipment [22,23,24] is available, it is typical to use generic equipment, such as general-purpose oscilloscopes and signal generators, to perform the tests. However, in this research, a self-developed SHM ultrasound system (SHMUS) was considered [25]. This system can perform ultrasound testing with up to 18 input/output channels simultaneously and can operate in both active and passive modes [26]. Here, the active mode was chosen, and sine waveforms of 300 kHz and 48 V peak-to-peak amplitude were programmed in this SHMUS to excite the PWAS and to be transmitted along the structure.

Figure 1 shows the block diagram of the self-developed SHMUS. The tool is based on a Field Programmable Gate Array (FPGA) integrated circuit. It shows an SHMUS that includes N channels, one channel per PWAS to generate and acquire acoustic waveforms. The generation of signals starts in the direct digital synthesis (DDS) block inside the FPGA. Next, the signal crosses the signal generation stage (Gen) to adapt the signal generated in the FPGA to the needs of the test. It is composed of a digital-to-analogue converter (DAC), a power circuit to deliver enough power to the PWAS, and a filter to clean the waveform provided to the transducers. 

After the propagation of the waveform along the structure, the waveform is received in the PWAS and the signal acquisition stage (Acq). It is composed of a conditioning circuit, a low noise operational amplifier (LNA), a filter, and an analogue-to-digital converter (ADC). After that, the digital signal reaches the pre-processor unit (PPU), which extracts meaningful information from the received waves (maximum and minimum points of the waves). The information is stored in FIFO memories. This SHMUS also includes the control unit, the power supply, and the data transfer connection by USB. The FPGA control unit transmits the acquired and pre-processed data to a computer. The red waveforms in Figure 1 show the emission of a sine waveform on channel 1 and the reception on channel 2 that occur in a simple test (named E1R2).

The setup for the tests includes the specimens under tests (plates made of aluminum and CFRP with PWAS bonded with epoxy), SHMUS, and a computer, which includes a self-developed software to control the performance of SHMUS and process the signals acquired (see Figure 2). 

Before starting the tests, the plates were scratched, and a hole was drilled in the aluminum plate (see upper part of Figure 2) to accelerate the appearance of corrosion. During the first week of the test period, the plates were kept apart to avoid corrosion and, during the subsequent weeks, they were in contact as shown in Figure 2.

There are numerous algorithms and techniques for UGWT signal processing [27,28,29]. These usually correspond to short time laboratory tests. In this research, the usefulness of several types of algorithms to detect progressive damage in three stages is demonstrated. Figure 3 shows the flowchart of the proposed experimental procedure from performing tests to data analysis.

The stages are designed according to the input data, the amount of data, and the time of execution. On the one hand, the digital circuits inside the FPGA run the first stage. On the other hand, the proprietary software in the control computer runs the second and third stages. The following three sections describe the three stages of processing. 

Before undertaking the signal pre-processing stage, it is necessary to analyze the way the tests were performed and to measure the repeatability and accuracy of the tests. The tests were performed in active mode using the round-robin mode. Namely, each test consisted of 10 consecutive emissions through the 10 PWAS and the subsequent acquisition of signals in all of the PWAS, which yielded 100 signals in each test (10 emissions multiplied by 10 acquisitions). In fact, since waveforms were acquired with all the PWAS in each test, it can be considered that the tests were performed using fast round-robin mode, in contrast to an ordinary round-robin test, where 100 simple tests would have been conducted. 

When acoustic waves are emitted from a single PWAS, the reflected and acquired waves are the composition or interference of waves that propagate in all directions and reflect off obstacles or edges. As the waves propagate along the structure, they suffer constructive and destructive interferences, and thus some areas of the structure may not receive the wave front. Therefore, the acquired waves may not contain information from these zones either. By using multiple emitters placed in different positions, the unmonitored zones change and with the appropriate number of emitters, the entire surface of the structure can be considered as monitored.

Figure 4 shows the signals emitted by PWAS 1 and received by PWAS 2 to 9, i.e., signals E1R2 to E1R9. The horizontal axis represents the time-of-flight (ToF) of the signal from the beginning of the test. The vertical axis shows the amplitude of the signal in mV. Even though all the signals are acquired in the same test, their waveforms are different, i.e., they show different amplitudes for the same ToF. 

To evaluate the test repeatability, all of them were performed three times, and the coincidence was checked periodically. Figure 5 shows three signals corresponding to the same day, material, and transmitter-receiver pair. The signals coincide largely and show only small differences. The time accuracy of SHMUS is 16.6 ns, and the amplitude accuracy 0.28 mV, but the signal dispersion is about 50 ns in the ToF measurement and 3 mV in the amplitude (top of Figure 5). 

On the one hand, the dispersion in time measurements coincides with the measurement precision plus the digitization error. Thus, ToF can be used to determine the behavior of the waves as they propagate through the structure and, consequently, to monitor the health of the structure. 

On the other hand, the amplitude dispersion in the measurements exceeds the amplitude accuracy of SHMUS by far. Even though signals are acquired in differential mode, the voltage drops may be affected by low frequency interference not related to the test. Thus, the amplitude is considered to determine the ToF of the maximums and minimums of the signal, but its value may not represent the state of health of the structure. 

## 3. Hardware Pre-Processing Algorithms

The first stage of each channel receives the signal acquired through the PWAS and Acq circuits. All channels process the hardware algorithm concurrently. To the best of the authors’ knowledge, the literature presents the signal processing algorithms run by computers. However, here the realization of a first signal pre-processing stage using logic circuitry designed within the FPGA is proposed. Monitoring an aircraft with hundreds of structures several times a day requires a large amount of memory and processing time. Signal processing by hardware means reduces the amount of data and processing time significantly, as shown below. 

The tests in the research last approximately 2 ms, at 60 MSPS, and this means 128,000 samples acquired per signal. A round-robin test with 10 PWAS produces 100 signals per test. Then, each monitoring process generates 12,800,000 samples for each structure; these data must be further processed. The pre-processing proposal introduced in this paper consists of obtaining the characteristic points (CP) of each signal, i.e., filtering the signal and calculating the maxima and minima that exceed a threshold voltage. Each CP is represented by the signal amplitude (Amp) and the ToF, resulting in a series of 64 pairs (Amp and ToF) per signal. This algorithm was implemented following the proposals of Castillero [30] and Gil-Garcia [31]. 

Figure 6 compares a regular acquired signal (blue line) and the CPs extracted from it (set of brown points). The set of CPs represent the most significant aspects of the signal, and therefore the main frequency, ToF of groups, most significant amplitudes, etc. can be calculated from the CPs. The set of CPs requires 1000-times less memory than the original signal and take much less time to be processed. Other signal features, such as zero crossings, maximum slopes, and the FFT, can also be calculated. The FPGA concurrently performs the computation of CPs for all channels during acquisition, and thus the processing time cost is zero.

The damage produced by corrosion was the type of damage chosen for the laboratory testing. Corrosion may happen in any part of the structure when the structure is not properly protected. Moreover, this is common damage to many materials exposed to open air and even more so in marine environments. To analyze corrosion, a testing campaign was conducted on working days for eight weeks. During the first week, the structures were separated, and during the weeks under corrosion, the aluminum plate was placed on top of the CFRP plate. No significant variation in the acquired signals was observed after separating or joining the plates under test and vice versa. 

Figure 7 shows some of the signals acquired when monitoring the aluminum plate. These signals correspond to the emission with PWAS 1 and the reception with PWAS 4 (E1R4) performed on the first day of the week during the eight weeks of the testing campaign. The thick dots correspond to the CPs, and the lines correspond to the interpolation of the dots to ease the viewing of the signals after the pre-processing stage. Only CPs are shown in the detailed view without interpolation lines. 

During the testing campaign, 2400 tests were performed and 24,000 signals were acquired. They correspond to 40 days of testing on two plates. Note that three equal tests were conducted each day to ensure repeatability. Each round-robin test consisted of 10 single tests, and 10 signals are acquired in each test. Figure 7 only represents the signal of one day of the week, during the eight weeks of tests. The detailed view in Figure 7 shows small variations in the ToF of the CPs.

Despite the fact that several days go by from test to test and corrosion is induced, the waveforms acquired show similar behavior. There are differences in the CPs obtained between 60 and 80 µs; however, their amplitude is not significant enough. The visual analysis of Figure 7 does not ease the identification of waveform patterns that can be associated with corrosion. Therefore, the graphical analysis of the signals is not sufficient to determine the state of the structure. In addition, the amount of data to be processed and the complexity of the measurements require signal-processing algorithms.

## 4. Pattern-Matching Process

The signal processing algorithms used in SHM usually are algorithms for comparison with a pristine signal (*pattern-matching*) or algorithms for analyzing signals for patterns (*pattern-recognition*) [32]. In this research, two algorithms, one of each type, arranged in consecutive stages were included.

First, as the second stage in Figure 3 a *pattern-matching* algorithm was considered. For this purpose, on the first day of the test campaign, the pristine state pattern was generated. In the initial health state of the structure, 10 tests were performed on each structure, generating 100 signals per trial. CPs with a ToF difference below a threshold were identified in each set of 10 signals, which correspond to the same transmitter–receiver pair. These CP correspond to the maximum and minimum amplitude points of the waveforms acquired with the PWAS in pristine state. 

Pristine CPs that were not identified in all these initial tests were removed as they are not considered meaningful. Next, a trapezoidal fuzzy set was generated for each pristine CP following the fundamentals of fuzzy logic [33,34,35] to determine the degree of membership (µ∈[0,1]). Fuzzy sets represent the possibility of finding a CP on a certain time interval. The equations that define the trapezoidal fuzzy set are given in Equation (1).
(1)$$µ=\{\begin{array}{cc} 0\;, & \;|ToF<\min\left(ToF_{i}\right)-\alpha \cdot T\\ \frac{\mathrm{ToF}-[\min\left(ToF_{i}\right)-\alpha \cdot T]}{\left(\alpha -\beta \right)\cdot T}, & |\min\left(ToF_{i}\right)-\alpha \cdot T\leq ToF<\min\left(ToF_{i}\right)-\beta \cdot T\\ \;1\;, & |\min\left(ToF_{i}\right)-\beta \cdot T\leq ToF\leq \max\left(ToF_{i}\right)+\beta \cdot T\\ \frac{[\max\left(ToF_{i}\right)+\alpha \cdot T]-ToF}{\left(\alpha -\beta \right)\cdot T}, & |\max\left(ToF_{i}\right)+\beta \cdot T<ToF\leq \max\left(ToF_{i}\right)+\alpha \cdot T\\ 0\;, & \;\;|\max\left(ToF_{i}\right)+\alpha \cdot T<ToF \end{array}$$
where $T=\max\left(ToF_{i}\right)-\min\left(ToF_{i}\right)$ and the coefficients α and β are two values to adjust the similarity range (the amplitude and slope of the trapezoid). The values considered were α=0.15 and β=0.1. Figure 8 shows the CPs of a pristine signal and the fuzzy sets associated to the signal.

In subsequent tests, the CPs of each acquired signal were identified and compared to the fuzzy sets associated with their pristine signal. The result of the comparisons is the degree of membership (*µ_i_*) of each CP according to Equation (1). The algorithm is detailed in [36]. Then, the Degree of Identity (DoI) of each signal ExRy (*DoI*_*x*,*y*_) was calculated with the mean of the *µ_i_* of its CPs (see Equation (2)).
(2)$$DoI_{x,y}=\frac{\sum_{i}\mu_{i}}{\#CP}$$

The *DoI*_*x*,*y*_ are then reversed and arranged in the Degree of Health (DoH) matrix of the structure, Equation (3). The DoH matrix is a 10 × 10 matrix representation that expresses the agreement of the signals acquired in the test with respect to the pristine state signals.
(3)$$DoH=\left(\begin{array}{ccc} 1-DoI_{1,1} & \cdots & 1-DoI_{1,10}\\ \vdots & \ddots & \vdots \\ 1-DoI_{10,1} & \cdots & 1-DoI_{10,10} \end{array}\right)$$

This algorithm requires very few computational resources. For example, in a round-robin test with 10 PWAS, the comparison of the 100 signals acquired, and the calculation of the DoH matrix takes less than one second. This fact is important for systems with hundreds of structures to be monitored several times a day. 

Figure 9 shows the DoH matrix from the aluminum structure obtained each Monday of the eight weeks of the test campaign. Each cell in each one of the eight matrices summarized shows the number obtained after the processing of the data gathered in the tests. Additionally, each cell is colored to ease the understanding of the matrices (green means no damage, and red means complete damage). The degradation of the aluminum structure under corrosion can be easily appreciated with the help of the number and color of each cell. As days go by, the values of various cells of the matrices increase. Furthermore, the colorful representation starts with all green cells, showing no damage in the pristine state, and evolves to red tones when corrosion damages the structure from week to week.

In each test, the PWAS receives complex signals. When a signal is emitted through a PWAS, the acoustic waveform propagates along the structure in all directions. During the propagation, the amplitude of the waveform is attenuated and reflections with obstacles or at edges of the structure happen. Furthermore, the signals acquired are combined in a nonlinear way in the algorithm of calculation of the DoH matrices. Thereby, the result is complex, and a theory that studies complex systems or Chaos Theory [37] must be taken into account. 

The damage caused to the structures under test is similar to what would happen in a real-world scenario. The pristine states correspond to the health states of the structures once they are at the customer’s facility, i.e., once they have suffered some damage and deterioration may be ongoing. Furthermore, the initial damage was extended over an undetermined part of the structures. The contact area between the two plates was not delimited. Moreover, on several occasions some salty water was added to promote corrosion without measuring the amount, salinity, or position. All this lack of definition in the test campaign brings the tests closer to the complexity that exists in a real-world environment. 

In summary, rather than a detailed measurement, a trend of the deterioration of the structure was obtained from the pattern-matching processing. A summarized or compressed way of assessing the health state of the structure at a given time was achieved. However, since the damage to be detected is progressive, an algorithm for interpreting the DoH matrices over the days of the test campaign is required.

## 5. Pattern-Recognition Process

The third stage of processing is based on an algorithm to identify patterns. *Pattern-recognition* is a process of categorizing samples of measurements or observed data as members of a class or category [38]. Several techniques are used to build the pattern-recognition stage, including neural networks, combined with more advanced techniques, such as chaos theory, decision trees, and genetic algorithms among others [39,40]. There is also a wide set of statistical algorithms that can be used in this stage. In this research, different types of means, median, and standard deviation that look for trends compatible with corrosion damage were implemented. The previous work performed by the pre-processing hardware algorithms and, especially, the pattern matching process, simplifies the data processing. Therefore, the computational cost in this stage is reduced. 

The input data for this stage are the DoH matrices of each day of the test campaign. These matrices summarize, into 10 × 10 data, the structure health of each day. It seems appropriate to use statistical algorithms at this stage because the signals have already been processed, discrete data are available, and the processing is intended to be computationally inexpensive. The previous stages analyzed a moment in time; however, this stage considers time as an input variable because it relates over time all the previously calculated DoH matrices. 

The four most common means in statistics are the arithmetic, geometric, harmonic, and quadratic mean. All these means are statistics of central tendency and they are suitable for analysis of the representation of a set of data coming from a homogeneous composition. An important criterion for the correct use of the mean is the determination of the scale to be used. According to Stevens [41], there are four types of scales: nominal, oral, interval, and ratio. In this case, the ratio scale is the suitable one, because the results show differences among the elements of the DoH matrix, and there is an absolute zero, that is, the total absence of the characteristic (in this case, absence of damage). This zero, in monitoring of structures, would mean a very correct state of the structure.

The most commonly used mean is the *arithmetic mean*. This is more appropriate to use when the distribution is symmetrical or approximately symmetrical; when an inferential analysis is desired or other statistics are to be used, such as the standard deviation or the correlation coefficient; when the scales of the data are interval or ratio; and when the distribution of the data is uniform. It ceases to be a representative value of the sample when the data present much variability or dispersion—that is, when it is affected by extreme data. Equation (4) shows the arithmetic mean of the matrices DoH for one day of the test campaign.
(4)$$\bar{x}=\frac{DoI_{1,1}+DoI_{1,2}+\ldots +DoI_{10,10}}{100}=\frac{\sum_{k,l=1}^{10,10}DoI_{k,l}}{100}$$

Regarding the *geometric mean* (Equation (5)), the data set must be positive numbers, and it is recommended to give greater importance to small values, when the data have a geometric or percentage growth, when index numbers or financial or accounting ratios are to be averaged, or when the data have to be used in terms of their logarithms. The geometric mean is affected by extreme values to a lower degree than the arithmetic mean.
(5)$${\bar{x}}_{g}=\sqrt[{100}]{DoI_{1,1}\cdot DoI_{1,2}\cdot \ldots \cdot DoI_{10,10}}$$

The *harmonic mean* (Equation (6)) of a data set is the inverse of the arithmetic mean of the inverses of these data. It is preferably used to calculate the velocity averages (the average velocity at equal spacing; as opposed to the average velocity at equal times, for which the arithmetic mean would be used). In general, it is useful when there are changes in the state of the variable (such as the flow rates or data or signal-reception rates). It is even less affected by extreme values than the geometric mean. However, it is sensitive to much smaller values than the set.
(6)$${\bar{x}}_{h}=\frac{100}{\frac{1}{DoI_{1,1}}+\frac{1}{DoI_{1,2}}+\ldots +\frac{1}{DoI_{10,10}}}$$

The *quadratic mean* (Equation (7)) is the square root of the arithmetic mean of the squares of the data values, and it is usually influenced by extreme values, especially large ones.
(7)$${\bar{x}}_{q}=\sqrt{\frac{DoI_{1,1}^{2}+DoI_{1,2}^{2}+\ldots +DoI_{10,10}^{2}}{100}}$$

In addition, the *truncated (or trimmed) mean* can also be used when extreme values are present and it is desired to eliminate their effect on the analysis because they may not be representative, as for example in the implementation of the SHM. This mean is defined as the arithmetic mean of the resulting data after excluding from the top *n*% of the data and from the bottom *n*% of the data. In this case, *n* = 10% was used, that is, 20% is excluded (10% from the top part and 10% from the bottom part). Since the truncated mean excludes extreme values, its usual purpose is to avoid the distortion that extreme scores may cause.

The *median* is the average value below which 50% of the data are found and can be advantageous when the distribution of the data is asymmetric; when there are extreme values that would distort the meaning of mean; for fuzzy values—that is, when there are distributions with undetermined values. The *standard deviation* (Equation (8)) was also used. It is defined as the square root of the arithmetic mean of the squares of the deviations from the mean, and it is a typical measure of the dispersion of values.
(8)$$\sigma =\sqrt{\frac{\sum_{k,l=1}^{10,10}{\left(DoI_{k,l}-\bar{x}\right)}^{2}}{100}}$$

Figure 10 shows all these statistical values applied to the DoH matrices representing the state of the aluminum structure during the test campaign. Here, these statistics show an increase, but not a linear increase; therefore, the circumstances that modify this trend should be analyzed. In particular, the temperature during the tests modifies the speed of wave propagation through the structure. If wave propagation has already been described as complex or chaotic, temperature changes makes it even more complicated. 

The calculation of each DoI depends on 100 signals. The signals acquired in the SHM tests are composed of waveforms propagated along the material in all directions as well as the reflections of the waveforms in the ends of the surface and the irregularities that may appear. Therefore, each signal is multiply dependent on temperature, and it is necessary to measure and analyze the effect of temperature during monitoring. 

During the test campaign, the temperature on the surface of the structure was measured with a Flir i7.0 thermal imaging camera that was placed on top of the monitored structure. Figure 11 shows the arithmetic mean of the DoH matrices on the aluminum plate during the test campaign and the temperature of each day. A certain proportionality is visible at first glance, and thus it is necessary to determine if the trend in the mean is due exclusively to temperature or if it also corresponds to corrosion damage. 

To analyze the result, a horizontal line at 22 °C and a series of vertical lines on days when the temperature was close to 22 °C are drawn on Figure 11. The black arrow, which determines the trend of the damage, joins points of the arithmetic mean statistic that were acquired at 22 °C. For example, for the values for days or samples 7 and 25, where the temperature was 22 °C, the numerical value changes from 0.084 to 0.148. Therefore, in spite of the temperature differences, the arithmetic mean reflects a deterioration of the structure with respect to its pristine state. If the rest of the statistics are analyzed, the same trend can be seen in all of them, i.e., how damage (due to corrosion) increases with respect to the pristine state as time progresses, which reinforces the proposed methodology to detect progressive damage in the structure.

## 6. Conclusions

This paper introduced a new methodology for detecting progressive damage in structures with ultrasonic guided waves. We detailed the complete setup for the tests. This consisted of self-designed equipment for the emission and acquisition of ultrasound waveforms through PWAS. A three-stage signal analysis procedure and the corresponding algorithms were also presented.

Progressive damage, such as corrosion, takes time to be noticeable. An eight-week test campaign was conducted, during which two structures, one made of aluminum and the other one of CFRP, underwent corrosion. The paper demonstrated that the methodology introduced allowed the detection of a trend in the deterioration of the structures. The test campaign demonstrated the suitability of the selected methodological options. Specifically: -The selection of 10 PWAS and the round-robin technique made it possible to analyze the structure based on 100 different signals. In this way, more information on the structure was obtained than by using a single PWAS.-The dedicated equipment made it possible to perform fast round-robin tests automatically, i.e., with no need to change connections during the test campaign.-The use of a first stage of hardware signal processing made it possible to reduce the amount of data to be transmitted and processed, as well as the total processing time. Specifically, due to the first stage of hardware signal processing, the amount of data to be processed was reduced by 1000.-The combination of the first two processing stages, round-robin technique, and SHMUS, provide a series of DoH arrays in only one second. The reduction of the acquisition and processing time is a key issue when attempting to scale this system to complex structures, such as those of an airplane.-The *pattern-matching* processing stage, which generates the DoH matrices, provided a reduced data set of 100 values that indicates the state of the structure whenever a test was conducted, for each day.-The results provided by the *pattern-recognition* processing stage showed a degradation trend of the aluminum structure that would allow the generation of maintenance orders.-The remarkable influence of temperature on the propagation velocity of ultrasound guided waves was shown.-The results obtained through DoH matrices and statistical parameters were shown to be nonlinear.-The complexity of the propagation of guided waves through a structure, typical of a chaotic system, was balanced with fuzzy and statistical algorithms that quickly provide a warning about the deterioration of the health of the structure.

This research could be continued with long-term tests for systems-on-board on aircraft. An issue of interest is to determine the optimal number of PWAS to perform the detection so that the hardware used in SHMUS can be reduced to the minimum necessary, thereby, reducing the cost and volume. An additional future task is the improvement of the algorithms and the search for a way to compensate for variations due to temperature change. The temperature range in airplanes is wider than in regular laboratories and so are the variations in the acquired signals while monitoring the health state of the structures. 

After optimizing all the issues described above and installing the systems on multiple structures, artificial intelligence and machine-learning algorithms could be included to complete the damage identification and localization process. 

## Figures and Tables

**Figure 1 sensors-22-01692-f001:**
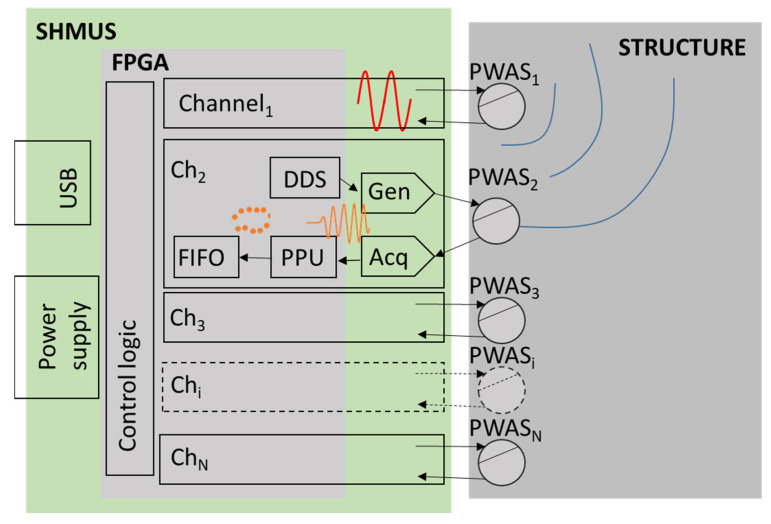
Block diagram of the self-developed SHMUS included in the tests (in red, situation for E1R2, i.e., emission with PWAS_1_ and reception with PWAS_2_).

**Figure 2 sensors-22-01692-f002:**
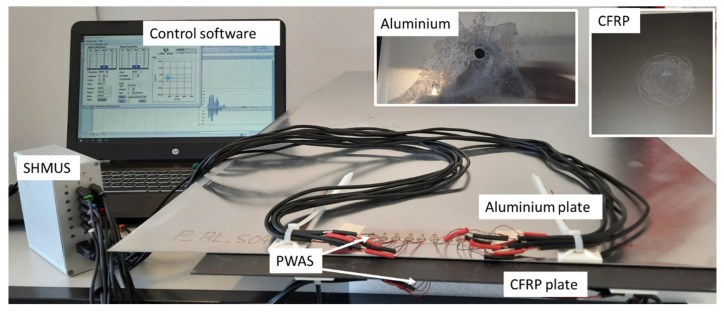
Complete setup for the tests on aluminum and CFRP specimens.

**Figure 3 sensors-22-01692-f003:**
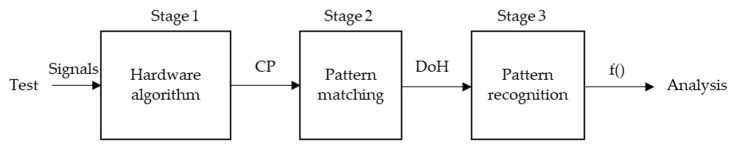
Flowchart of the experimental procedure proposal for detecting progressive damage in SHM monitoring.

**Figure 4 sensors-22-01692-f004:**
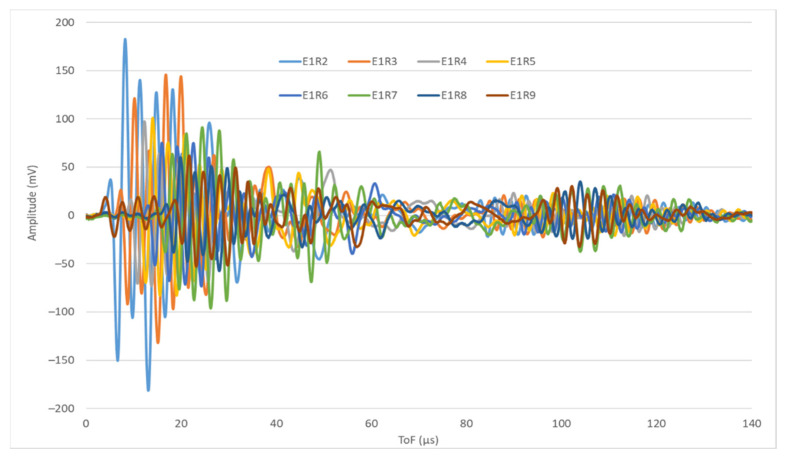
Signals E1R2-E1R9, i.e., signals emitted by PWAS 1 and received by PWAS 2 to PWAS 9.

**Figure 5 sensors-22-01692-f005:**
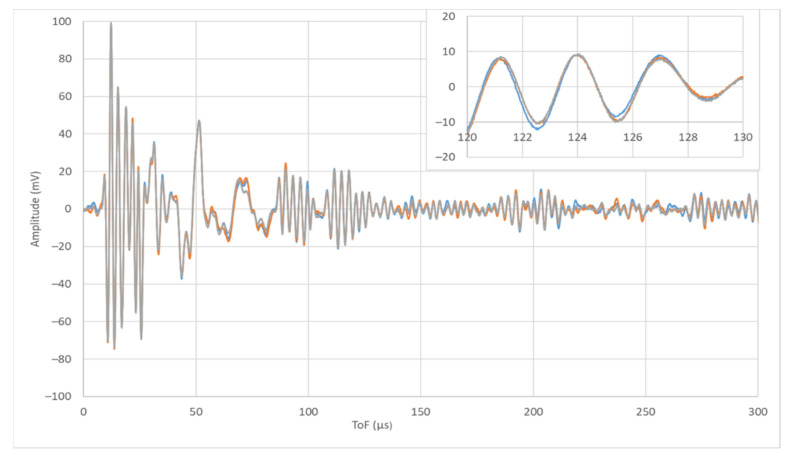
Repeatability of the ultrasound tests and dispersion of the acquired signals (50 ns and 3 mV). General and detailed views.

**Figure 6 sensors-22-01692-f006:**
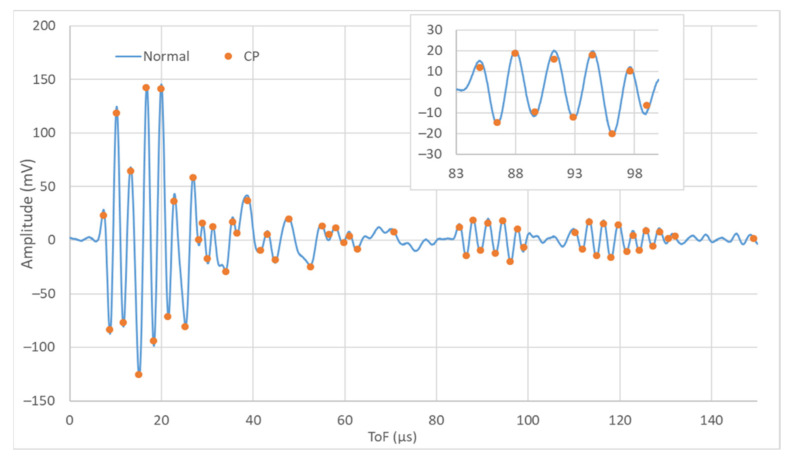
Comparison between a regular acquired signal and a pre-processed signal shown through their CPs (general and detailed views).

**Figure 7 sensors-22-01692-f007:**
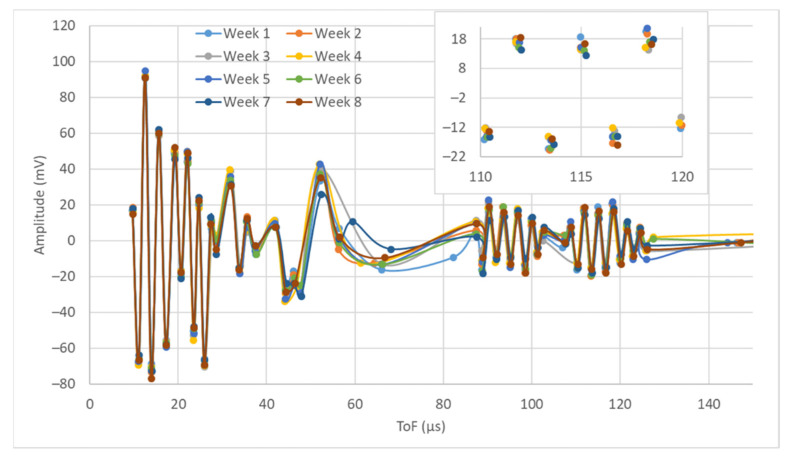
E1R4 signal acquired during the eight Mondays of the testing campaign. General view with CPs and interpolation lines and a detailed view with CPs.

**Figure 8 sensors-22-01692-f008:**
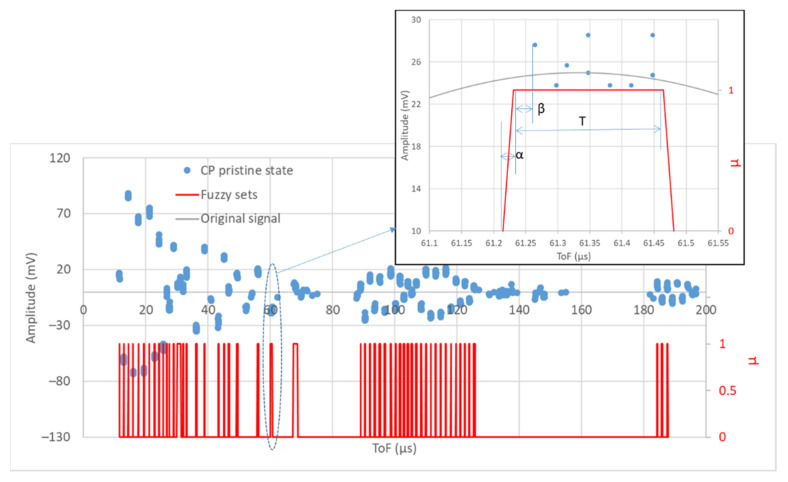
Pristine signal shown through its CPs and the fuzzy sets generated. General view and detailed view.

**Figure 9 sensors-22-01692-f009:**
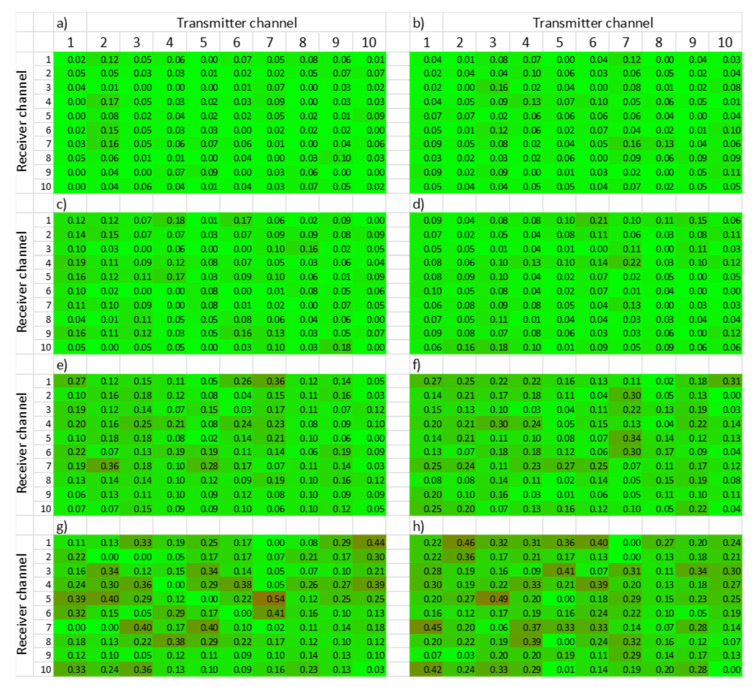
DoH matrices from the aluminum structure obtained on a weekly basis during the eight weeks of the test campaign. (**a**–**h**) weeks 1–8.

**Figure 10 sensors-22-01692-f010:**
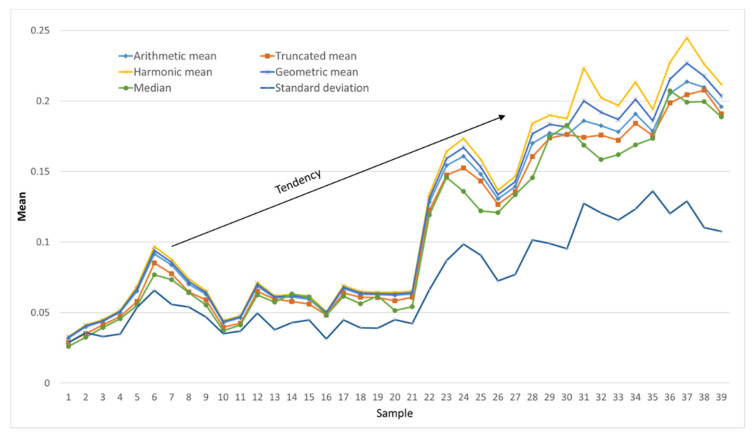
The means, median, and standard deviation calculated for the aluminum plate during the test campaign.

**Figure 11 sensors-22-01692-f011:**
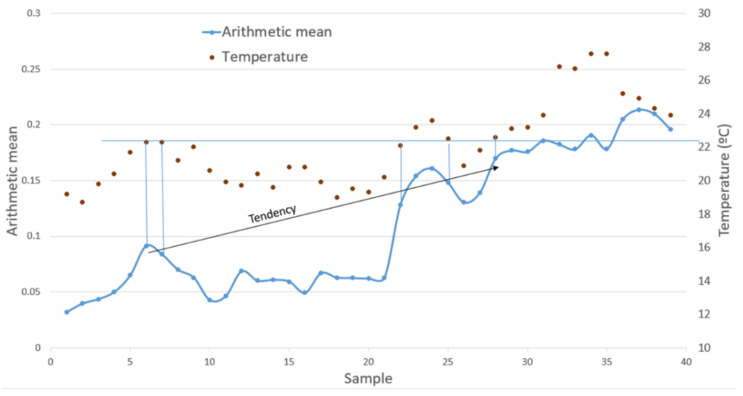
The arithmetic mean vs. temperature in the aluminum plate during the test campaign. Graphical analysis of the variation.

## Data Availability

Not applicable.

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
