# Peer review of "Methodology for Detecting Progressive Damage in Structures Using Ultrasound-Guided Waves"

_sensors, 2022, doi:10.3390/s22041692_

Round 1
Reviewer 1 Report
The manuscript needs substantial revision before it can be considered for publication. 1. The authors stated that "These materials are never put together, since aluminum in contact with CFRP produces a significant deterioration in both materials, especially under a marine environment". Why did the authors select a specimen of aluminum plate contact with CFRP plate? 2. Is there any way to optimize the number of sensors? For such a small plate, it seems that 10 sensors are more than enough. 3. How to quantitatively define the material deterioration by corrosion? Otherwise, the monitoring can hardly be used in practice.Author Response
Please see the attachment.

Reviewer 2 Report
The present paper deals with the analysis of progressive damage of structures by means of guided waves.
As a whole, the article presents an interesting methodology for damages evaluation of engineering structures, however I highlighted some points that must be reviewed.
>> The formal writing and English grammar must be fully revised in the document.
>> Line 43. There are several references that uses vibration techniques for structure monitoring. Some must be included in the text.
>> Line 50-57. The authors comment about the difficulty of using acoustic and electromagnetic techniques in CFRP and later in the same paragraph suggests the use of SHM with such techniques. The paragraph was confused. You should review the sentence and reference other works that use SHM in composites.
>> Formal writing is required, therefore, a global review in the text is necessary. Some examples of colloquialism:
Line 82: “The rest of the paper is structured as follows”. Formal writing is required.
Line 89: “we hypothesize the needs to perform the”. Formal writing is required.
>> fig 9. The way it is presented could be modified. Too many data in the table.
>> fig 11. The trend graphic for the CFRP sample is missing in the paper.
Author Response
Thank you so much for your kind review. We really appreciate your time for reading and analyzing it. Please see the attachment.

Round 2
Reviewer 1 Report
The reviewer has no further comments.
Author Response
Dear reviewer,
Thank you very much for the review. Definitely, it increased the quality of our manuscript.
Best regards,
The authors